# Epidemiology of canine ehrlichiosis and molecular characterization of *Erhlichia canis* in Bangladeshi pet dogs

**Ajran Kabir**[1⊛], **Chandra Shaker Chouhan**[2⊛], **Tasmia Habib**[1], **Md. Zawad Hossain**[3], **Abu Raihan**[3], **Farzana Yeasmin**[2], **Mahbubul Pratik Siddique**[1], **A. K. M. Anisur Rahman**[2], **Azimun Nahar**[2], **Md. Siddiqur Rahman**[2], **Md. Amimul Ehsan**[2]*

1 Department of Microbiology & Hygiene, Bangladesh Agricultural University, Mymensingh, Bangladesh,
2 Department of Medicine, Bangladesh Agricultural University, Mymensingh, Bangladesh, 3 Doctor of Veterinary Medicine, Bangladesh Agricultural University, Mymensingh, Bangladesh

⊛ These authors contributed equally to this work.
* amimul.med@bau.edu.bd

**Data Availability Statement:** All relevant data are within the manuscript and its Supporting Information files. The submission contains all the raw data which can be replicated. Raw data is

## Abstract

### Background

*Ehrlichia canis*, a rickettsial organism, is responsible for causing ehrlichiosis, a tick-borne disease affecting dogs.

### Objectives

This study aimed to estimate ehrlichiosis prevalence and identify associated risk factors in pet dogs.

### Methods

A total of 246 peripheral blood samples were purposively collected from pet dogs in Dhaka, Mymensingh, and Rajshahi districts between December 2018 and December 2020. Risk factor data were obtained through face-to-face interviews with dog owners using a pre-structured questionnaire. Multivariable logistic regression analysis identified risk factors. Polymerase chain reaction targeting the 16S rRNA gene confirmed *Ehrlichia* spp. PCR results were further validated by sequencing.

### Results

The prevalence and case fatality of ehrlichiosis were 6.9% and 47.1%, respectively. Dogs in rural areas had 5.8 times higher odds of ehrlichiosis (odd ratio, OR: 5.84; 95% CI: 1.72–19.89) compared to urban areas. Dogs with access to other dogs had 5.14 times higher odds of ehrlichiosis (OR: 5.14; 95% CI: 1.63–16.27) than those without such access. Similarly, irregularly treated dogs with ectoparasitic drugs had 4.01 times higher odds of ehrlichiosis (OR: 4.01; 95% CI: 1.17–14.14) compared to regularly treated dogs. The presence of ticks on dogs increased ehrlichiosis odds nearly by 3 times (OR: 3.02; 95% CI: 1.02–8.97). Phylogenetic analysis, based on 17 commercially sequenced isolates, showed different

submitted as supporting information under the file name "Minimal Dataset"

**Funding:** This project was funded by Bangladesh Agricultural University Research System (BAURES) (Project No. 2018/557/AU-GC). the funders were not involved in the study design, data collection and analysis, decision to publish, or the preparation of the manuscript.

**Competing interests:** All authors declare that they have no conflicts of interest.

clusters of aggregation, however, BAUMAH-13 (PP321265) perfectly settled with a China isolate (OK667945), similarly, BAUMAH-05 (PP321257) with Greece isolate (MN922610), BAUMAH-16 (PP321268) with Italian isolate (KX180945), and BAUMAH-07 (PP321259) with Thailand isolate (OP164610).

## Conclusions

Pet owners and veterinarians in rural areas should be vigilant in monitoring dogs for ticks and ensuring proper preventive care. Limiting access to other dogs in high-risk areas can help mitigate disease spread. Tick prevention measures and regular treatment with ectoparasitic drugs will reduce the risk of ehrlichiosis in dogs. The observed genetic similarity of the Bangladeshi *Ehrlichia canis* strain highlights the need for ongoing surveillance and research to develop effective control and prevention strategies, both within Bangladesh and globally.

## 1. Introduction

*Ehrlichia canis* (*E. canis*) is one of the most critical and deadly species within the genus *Ehrlichia*. It causes canine monocytic ehrlichiosis (CME), also known as tropical canine pancytopenia (TCP) [1]. *Ehrlichia canis*, a worldwide distributed rickettsial organism, is primarily transmitted by ticks, with the brown dog-tick (*Rhipicephalus sanguineus*) being globally responsible vector [2–7]. However, in East Asian countries, *Haemaphysalis longicornis* and *H. flava* are responsible for ehrlichiosis in dogs [2, 8]. *Ehrlichia canis* is an obligate intracellular, gram-negative, and extremely pleomorphic organism enveloped with a crinkled delicate outer membrane [9]. Notably, *Ehrlichiosis* is significant as a zoonotic disease, as *E. canis* is closely related to *Ehrlichia chaffeensis*, which causes human monocytic ehrlichiosis (HME) [10]. The clinical manifestations of CME caused by *E. canis* are strikingly similar to those of HME [10]. HME typically manifests as a mild disease in the vast majority of cases, whereas immunocompromised individuals experience the severe form of the disease [10].

Ehrlichiosis presents in three phases, including acute, subacute, and chronic [11]. During the acute phase, dogs may exhibit anorexia, depression, fever, lymphadenopathy, dyspnea, and slight weight loss. The chronic phase is the most severe and is characterized by emaciation, peripheral oedema, epistaxis, hemorrhages, and hypotensive shock, often leading to death [11]. *Ehrlichia canis* targets monocytes in the peripheral bloodstream and can be detected in these cells. The timely and proper administration of doxycycline or other tetracyclines is instrumental in facilitating the recovery of most dogs, whether in the acute phase or experiencing subclinical manifestations [11, 12]. However, a few dogs may progress to the chronic phase with a poor prognosis [13]. As ehrlichiosis is a vector-borne zoonotic disease, its incidence is linked to the prevalence and activity of ticks. Therefore, infections are mainly found in the warm season [11].

The diagnosis of ehrlichiosis can be challenging due to its distinct phases and diverse clinical presentations [11]. Various diagnostic methods can be employed, including microscopic examination of blood smears, serologic tests, and molecular techniques [14]. *Ehrlichia* spp. multiplying within membrane-bound vacuoles known as morulae can sometimes be observed through light microscopic inspection of stained blood smears. *Ehrlichia canis* and *E. chaffeensis* are found in monocytes in the stained blood smears, but *E. ewingii* is found in granulocytes. Both the immunofluorescent assay (IFA) and the enzyme-linked immunosorbent assay

(ELISA) are utilized to detect antibodies [15]. However, it is essential to note that there might be cross-reactivity of antibodies to different *Ehrlichia* species within the *Anaplasmataceae* family [16]. Polymerase chain reaction (PCR), a widely employed molecular technique, is pivotal in diagnosing *Ehrlichia* spp. infection, particularly in dogs exhibiting acute symptoms, as it can detect the disease before a detectable antibody response occurs [17].

The zoonotic importance of *E. canis* was historically overlooked until a groundbreaking study [10], reported human infections with clinical manifestations resembling human monocytic ehrlichiosis (HME). In South Asian countries, particularly Bangladesh, epidemiological knowledge about canine ehrlichiosis remains limited [2]. As a result, control and prevention efforts received little attention, posing a threat to both pet dogs and humans [17]. As the dog population in the country continues to grow, medical and veterinary personnel have further underestimated the risks to animal and human health. To bridge this knowledge gap and better understand *E. canis* infections in Bangladesh, this study aimed to conduct an epidemiological investigation and molecular characterization of this pathogen.

## 2. Methods and materials

### 2.1 Ethics statement

The samples were collected following the approved guidelines of the Committee of Experimental Operational Guidance and Animal Welfare at Bangladesh Agricultural University, under the permit number AWEEC/BAU/2019(51).

### 2.2 Study, target population, and sample size

Our study population consisted of pet dogs residing in both urban and rural areas and having regular contact with their owners. The target population for our study encompassed all domestic dogs in Bangladesh. The sample size was determined using the equation provided below:

$$n = \frac{1.96^2 PQ}{d^2}$$

In the above equation, 1.96 = Z value for the 95% confidence level, P = expected prevalence = 20% = 0.20 and Q = 1-P and d = precision = 5% = 0.05. We used 20% expected prevalence in the formula based on the average prevalence of ehrlichiosis in South Asia using PCR [2]. The assumptions used in the equation resulted in a sample size of 246.

### 2.3 Study area

This cross-sectional study was conducted in Bangladesh's Dhaka, Mymensingh, and Rajshahi districts (Fig 1). These areas represent two different climatic zones, with Rajshahi being the driest and hottest western zone and Dhaka and Mymensingh falling under the north central zone with mild summers and relatively heavy rainfall [18]. The weather conditions in these regions are conducive to the tick manifestation, which acts as a vector for this disease [19]. These specific areas were chosen based on their significant pet populations and popularity among pet owners. Samples were collected from different pet clinics within these selected regions, which we believe represent the pet dog populations in Bangladesh.

### 2.4 Sampling and data collection

The study purposively collected 246 blood samples from dogs between December 2018 and December 2020. Four milliliters of blood were drawn into an ethylenediamine tetra acetic acid (EDTA) tube from the saphenous vein. The collected samples were immediately transferred to

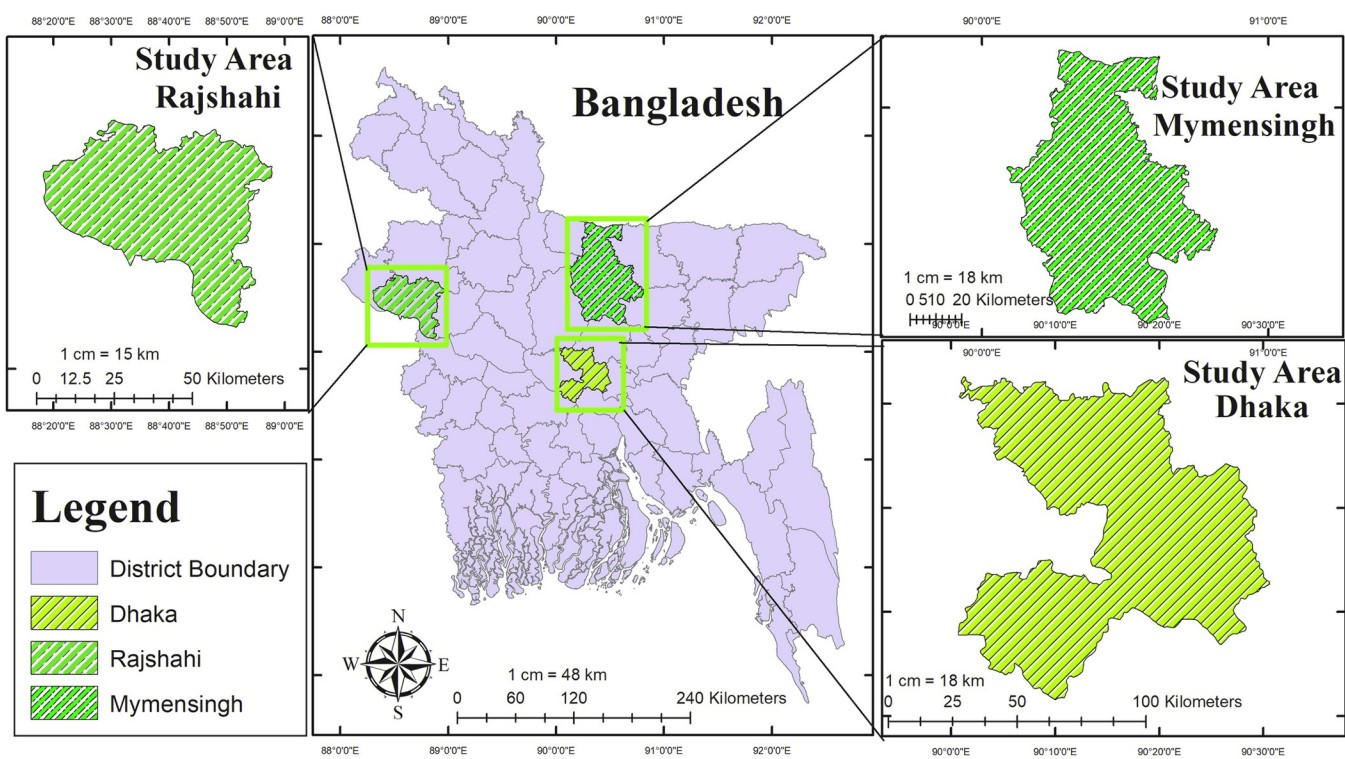

**Fig 1. Bangladesh map showing different sampling areas of *Ehrlichia canis*.** Note: This map was created by our team using ArcGIS version 10.1 (http://www.esri.com/arcgis).

the Department of Medicine, Faculty of Veterinary Science, Bangladesh Agricultural University, maintaining the cool chain.

During blood sampling, a questionnaire (S1 File) was filled out to gather data, including the dog's sex, age, breed, outdoor access, presence of ticks on the dog's body, regular ectoparasitic treatment (Owners who administer ectoparasitic treatments as per veterinary advice and within the specified period are categorized as "Yes," while those who deviate from this timeline are categorized as "No"), residence and other relevant risk factors associated with the spread of diseases [20].

## 2.5 DNA extraction and PCR

The Purelink™® DNA blood Mini kit (Invitrogen, USA) was employed to extract DNA from blood samples following the manufacturer's instructions [21]. The *Ehrlichia* DNA was amplified using a genus-specific array of *16S rRNA* gene primers. Specifically, the *Ehrlichia* agents were detected using the primer sets EHR16SD (5'-GGT-ACC-YAC-AGA-AGA-AGT-CC-3') and EHR16SR (5'-TAG-CAC-TCA-TCG-TTT-ACA-GC-3'). PCR was performed as previously described in the relevant studies [22, 23]. Commercially available Genomic DNA from *Ehrlichia chaffeensis*, strain St. Vincent (ATCC) was used as positive control for this genus specific PCR.

## 2.6 Sequencing and phylogenetic analysis

For partial sequencing, the PCR amplicons were purified using the FavorPrep™ GEL/PCR Purification Kit (FAVORGEN Biotech Corp., Taiwan) per the manufacturer's instructions.

The purified amplicons were then sequenced bidirectionally using an ABI 3500 Dx genetic analyzer (Applied Biosystems, USA) utilizing the commercial services of DNA Solution Ltd, Dhaka, Bangladesh. The obtained sequences were manually checked for potential sequencing errors in MEGA 11. Subsequently, they were used to query the GenBank online database nucleotide collection (nr/nt) using the megablast algorithm (optimized for highly similar sequences) in the basic local alignment search tool (BLAST) online application (https://blast.ncbi.nlm.nih.gov/Blast.cgi). Phylogenetic analysis was performed by constructing a neighbor-joining tree with evolutionary distances calculated using the Jukes-Cantor technique and a bootstrap value of 1000. The evolutionary analyses were conducted using MEGA 11.

### 2.7 Statistical analysis

The risk factors and *Ehrlichia* infection status data were entered and managed in a Microsoft Excel 2010 spreadsheet (S2 File) and later exported to SPSS 22 for statistical analysis. A dog was considered infected with *Ehrlichia* spp. if it tested positive by PCR. The age of dogs was transformed into a categorical variable. Case fatality was calculated through follow-up communication with the dog owners. Z-tests for proportions were used to compare prevalence and case fatality among districts [24, 25]. Initially, univariable logistic regression models assessed the relationship between *Ehrlichia* infection status and various explanatory variables. Multivariable logistic regression was conducted, considering explanatory variables with a p-value $\leq 0.2$. Multicollinearity among explanatory variables was checked by the variation inflation factor (VIF) with a cut-off of $\leq 5$. The final model was selected using a backward model selection technique, and model fit was evaluated using the Hosmer-Lemeshow test [26]. The potential confounding and interactions were also assessed using the previously described method [27].

## 3. Results

### 3.1 Prevalence and case fatality of ehrlichiosis

Out of 246 dogs tested, 17 cases were found to be PCR positive, resulting in a prevalence of 6.9%. The overall case fatality was 3.3% (Table 1). The prevalence and case fatality of ehrlichiosis were higher in the Mymensingh district (7.3%, 5.5%), followed by the Dhaka district (6.7%, 1.4%) and Rajshahi district (5.9%, 2.0%). Notably, the Mymensingh district showed the highest ehrlichiosis case fatality (66.7%), followed by the Rajshahi district (33.3%) and Dhaka district (20.0%). Fig 2 displays the PCR positive results, indicating the presence of *E. canis* DNA in these samples.

### 3.2 Univariable and multivariable analysis

The multivariable analysis included five factors associated with ehrlichiosis at a significance level of p$\leq$ 0.2 in the univariable analysis (Table 2).

**Table 1. Overall prevalence and case fatality of canine ehrlichiosis.**

| District | Tested | *Ehrlichia* spp. positive | Died | Prevalence (%) | Case fatality (%) |
|---|---|---|---|---|---|
| Dhaka | 72 | 5 | 1 | 6.9[a] | 20.0[a] |
| Mymensingh | 123 | 9 | 6 | 7.3[a] | 66.7[a] |
| Rajshahi | 51 | 3 | 1 | 5.9[a] | 33.3[a] |
| Overall | 246 | 17 | 8 | 6.9 | 47.1 |

[a]values with different superscripts within the same column for each variable don't differ significantly (p < 0.05).

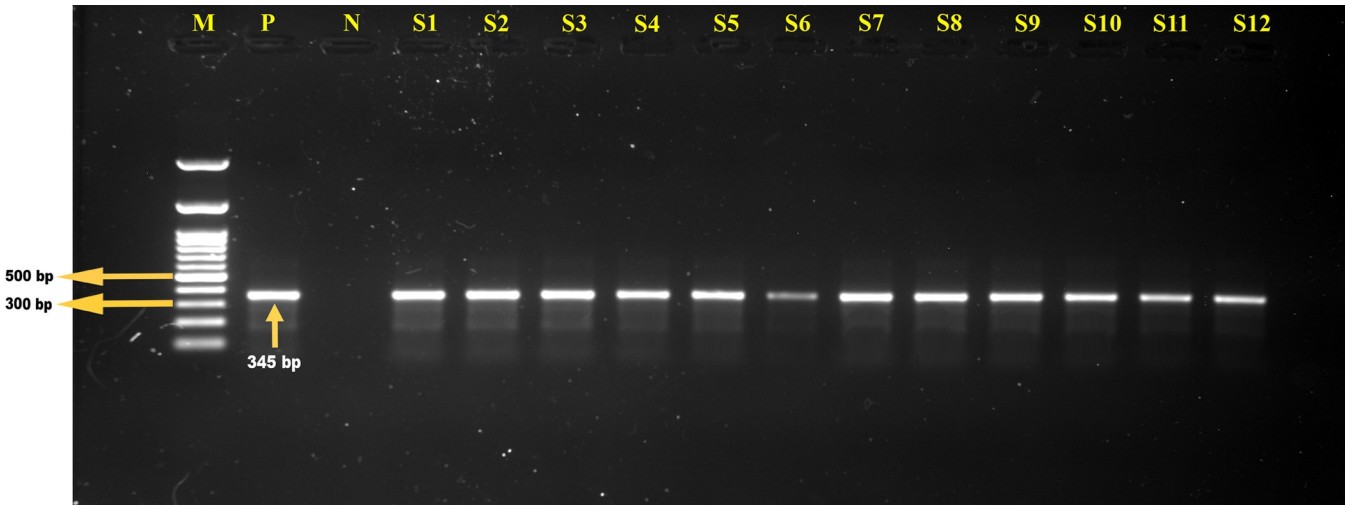

**Fig 2. Identification of *Ehrlichia spp.* by polymerase chain reaction.** Gel electrophoresis showing *16S rRNA* genes amplicons from *Ehrlichia* spp.. In Lanes: M- 100 bp DNA ladder (Promega, USA), P: Positive control, N: Negative control, and Lanes S1-S12: representing the samples of *Ehrlichia* spp. displaying bands of approximately 345 bp.

Dogs in rural areas had 5.8 times higher odds of ehrlichiosis (odd ratio, OR: 5.84; 95% CI: 1.72–19.89) than urban areas. Dogs with access to other dogs had 5.14 times higher odds of ehrlichiosis (OR: 5.14; 95% CI: 1.63–16.27) than those without such access. Similarly, irregularly treated dogs with ectoparasitic drugs had 4.01 times higher odds of ehrlichiosis (OR: 4.01; 95% CI: 1.17–14.14) compared to regularly treated dogs. Ticks on dogs increased ehrlichiosis odds nearly three times (OR: 3.02; 95% CI: 1.02–8.97) (Table 3).

**Table 2. Univariable association between *Ehrlichia canis* infection status and other explanatory variables.**

| Risk factors | Category level | OR | 95% CI | P-Value |
|---|---|---|---|---|
| Age (month) | ≤ 12 months | Ref | - | - |
| | 13 - ≤ 24 months | 1.80 | 0. 46–6.99 | 0.397 |
| | ≥ 25 months | 1.82 | 0.53–6.28 | 0.342 |
| Sex | Male | Ref | - | - |
| | Female | 1.56 | 0.58–4.19 | 0.379 |
| Breed* | Local | Ref | - | - |
| | Exotic | 2.82 | 1.03–7.70 | 0.043 |
| Presence of tick* | Yes | 4.24 | 1.55–11.60 | 0.005 |
| | No | Ref | - | - |
| Outdoor access* | No | Ref | - | - |
| | Yes | 4.08 | 1.45–11.47 | 0.008 |
| Regular ectoparasitic treatment* | No | 3.98 | 1.26–12.56 | 0.019 |
| | Yes | Ref | - | - |
| Season | Summer | 2.21 | 0.57–7.99 | 0.250 |
| | Rainy | 1.20 | 0.28–5.20 | 0.811 |
| | Winter | Ref | - | - |
| Residence* | Rural | 3.64 | 1.24–10.68 | 0.019 |
| | Urban | Ref | - | - |

OR: Odds ratio; CI: 95% confidence interval

* Candidate variable for multivariable analysis

Table 3. Risk factors for canine ehrlichiosis identified in the final multivariable logistic regression model.

| Risk factors | Category level | OR | 95% CI | P-Value |
|---|---|---|---|---|
| Presence of tick | Yes | 3.02 | 1.02–8.97 | 0.047 |
| | No | Ref | - | - |
| Outdoor access | No | Ref | - | - |
| | Yes | 5.84 | 1.72–19.89 | 0.005 |
| Regular ectoparasitic treatment | No | 4.01 | 1.17–14.14 | 0.027 |
| | Yes | Ref | - | - |
| Residence | Rural | 5.14 | 1.63–16.27 | 0.005 |
| | Urban | Ref | - | - |

OR: Odds ratio; CI: 95% confidence interval

### 3.3 Molecular characterization and phylogenetic analysis

The sequences of all PCR-positive amplicons were submitted to GenBank with the following accession numbers: PP321253 (BAUMAH-01), PP321254 (BAUMAH-02), PP321255 (BAU-MAH-03), PP321256 (BAUMAH-04), PP321257 (BAUMAH-05), PP321258 (BAUMAH-06), PP321259 (BAUMAH-07) PP321260 (BAUMAH-08) PP321261 (BAUMAH-09) PP321262 (BAUMAH-10) PP321263 (BAUMAH-11) PP321264 (BAUMAH-12) PP321265 (BAUMAH-13) PP321266 (BAUMAH-14) PP321267 (BAUMAH-15) PP321268 (BAUMAH-16) and PP321269 (BAUMAH-17). Among the 17 sequenced isolates, BAUMAH-13 (PP321265), BAUMAH-05 (PP321257), BAUMAH-16 (PP321268), and BAUMAH-07 (PP321259) exhibited 100% similarity with *Ehrlichia canis* isolates Foshan3-4 (OK667945) from China, VH1-58 (MN922610.1) from Greece, M66 (KX180945) from Italy, and K35 (OP164610) from Thailand, respectively. Moreover, these isolates were closely related phylogenetically (Fig 3). Notably, all the isolates were grouped into two separate and distinct clades.

## 4. Discussion

This study presents the first-ever description of the prevalence of canine ehrlichiosis in Bangladesh and identifies associated risk factors. Additionally, molecular characteristics of *E. canis* isolates were provided. To control canine ehrlichiosis, we recommend pet owners and veterinarians in rural areas monitor dogs for ticks and prioritize proper preventive care. Limiting access to other dogs in high-risk areas can be instrumental in curbing the spread of the disease. Implementing effective tick prevention measures and ensuring regular treatment with ectoparasitic drugs are essential to reduce the risk of ehrlichiosis in dogs.

The overall prevalence of canine ehrlichiosis in Bangladesh, as determined by conventional PCR in this current study, was 6.9%. This study's finding aligns closely with reported prevalence in other countries, such as Argentina at 6.7% [28], India at 8.0% [29], Egypt at 9.7% [30], and Paraguay at 10.41% [31]. However, a wide range of prevalence has been observed in various Southeast Asian countries, ranging from 0.39% to 86.90% in India [32, 33], 28% in Pakistan [20], and 8% to 27.14% in Nepal [20, 34]. Moreover, disparate prevalence levels have been reported in regions, including 2.0% in Malaysia [35], 20.41% in Turkey [36], 29.26% in Mexico [37], 32% in Costa Rica [38], 40.6% in Columbia [39], and 51.3% in Peru [11]. The variations of *E. canis* infection observed worldwide can be attributed to several factors, including the abundance and density of vectors, socioeconomic conditions, diverse climatic conditions, sample sources, and the diagnostic methods applied [2, 30, 31, 40]. The overall case fatality in this study was 47.1%. The higher case fatality observed in affected dogs may be due to lack of

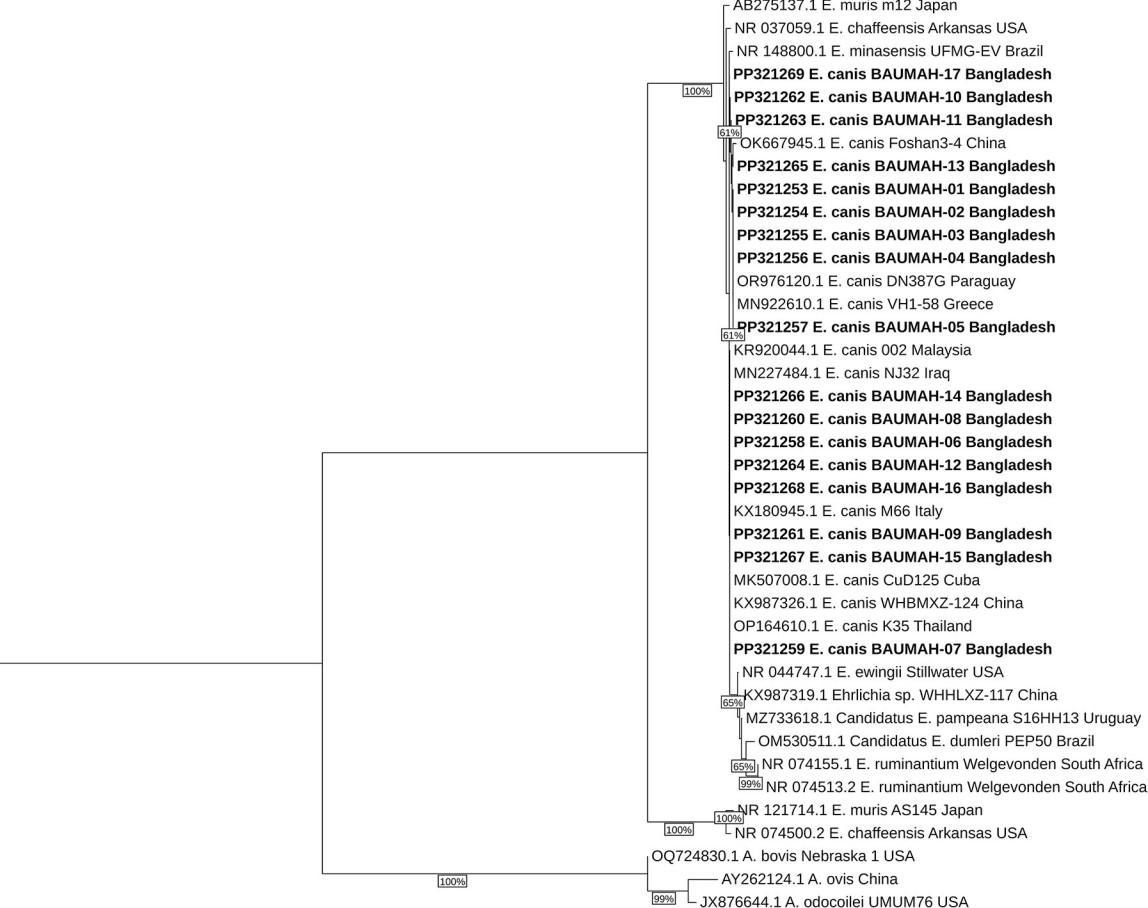

**Fig 3. Phylogenetic tree of *E. canis*.** The phylogenetic tree was constructed using 40 partial *16S rRNA* gene sequences of *E. canis* where 17 of them sequenced in this study. The Neighbor-Joining method was utilized for tree construction, and the evolutionary distances between sequences were calculated using the Tamura-Nei method. The analyses were performed using MEGA11 software, and the tree was visualized in iTOL. In the phylogenetic tree, the sequences obtained in this study are highlighted in red, indicating their placement and relationship with other *E. canis* sequences.

accurate diagnosis and proper treatment. In this study, prevalence was similar across districts, but the Mymensingh district had the highest case fatality. Most of the dogs that succumbed in the Mymensingh district were from border areas.

In the current study, regular use of ectoparasiticidal drugs was found to decrease the odds of ehrlichiosis in dogs significantly. These findings are consistent with previous studies [2, 30, 31]. The observed association between regular use of ectoparasiticidal drugs and reduced odds of ehrlichiosis could be attributed to ticks' essential roles in transmitting and spreading infection. Regularly using ectoparasiticidal drugs may help minimize *E. canis* infection by limiting tick transmission.

The presence of ticks on a dog's body was found to be significantly associated with the occurrence of ehrlichiosis. Ticks serve as the main arthropod vectors responsible for transmitting ehrlichiosis, and several studies have corroborated the role of various tick species in the

spread of the disease [2, 30, 31]. The efficient transmission of *E. canis* requires as little as 3 hours of tick adherence to dogs [31].

Dogs residing in rural areas had significantly higher odds of *E. canis* infection than dogs in urban areas. This finding aligns with a previous study by Carvalho, Wenceslau [41]. The increased odds in rural areas can be attributed to dogs in such settings often roaming outdoors, making them more susceptible to tick infestation [42]. In our country, rural dogs may not receive adequate veterinary care, hygiene, or regular ectoparasiticidal treatments, further contributing to their higher vulnerability to ehrlichiosis.

The sequence analysis of our isolates provided conclusive evidence of the presence of *E. canis* in Bangladeshi dogs, demonstrated their phylogenetic clustering and, interestingly, their genetic similarity to published isolates from China, Greece, Italy, and Thailand. Notably, the genetic diversity observed among our isolates confirms the existence of diverse strains of *E. canis* circulating in Bangladesh.

We collected from veterinary clinics rather than through household surveys. It is true that dog owners tend to seek veterinary assistance when their dogs are sick, which may lead to an overrepresentation of dogs with more severe signs, while those with mild signs or in a subclinical phase might not have been investigated. However, it is also important to note that dogs are brought to veterinary clinics not only for illness but also for deworming, vaccination, and routine health check-ups. The dog owners in our sample are generally educated and affluent, representing a segment of society that is particularly aware of their pets' health. Therefore, we believe our sample provides a representative snapshot of the pet dog population in Bangladesh. Nonetheless, it is a limitation of our study that only dogs visiting pet clinics were included. To explore the full burden of the disease, future studies should also consider stray dogs and those not routinely brought to clinics.

## 5. Conclusion

This study emphasizes the importance of proactive measures by pet owners and veterinarians in rural areas to combat canine ehrlichiosis. Vigilance in monitoring dogs for ticks and ensuring regular preventive care can significantly reduce the odds of disease. Limiting access to other dogs in high-risk areas can also play a crucial role in mitigating disease spread. Adopting tick prevention measures and regular treatment with ectoparasitic drugs is essential in safeguarding dogs from *E. canis* infection. Moreover, the genetic similarity observed between the Bangladeshi *E. canis* isolates and strains from other countries indicates the importance of continuous surveillance and research efforts. Understanding the genetic diversity and epidemiological characteristics of the disease on a global scale will aid in developing effective control and prevention strategies within Bangladesh and on an international level.

## Supporting information

**S1 File. Questionnaire used for data collection.**
(DOCX)

**S2 File. Dataset utilized for statistical analysis.**
(XLSX)

## Author Contributions

**Conceptualization:** Ajran Kabir, Chandra Shaker Chouhan, Md. Amimul Ehsan.

**Data curation:** Chandra Shaker Chouhan, Abu Raihan.

**Formal analysis:** Chandra Shaker Chouhan.

**Funding acquisition:** Md. Amimul Ehsan.

**Investigation:** Ajran Kabir, Tasmia Habib, Md. Zawad Hossain, Abu Raihan, Farzana Yeasmin.

**Methodology:** Ajran Kabir, Chandra Shaker Chouhan, Tasmia Habib, Md. Zawad Hossain.

**Software:** Ajran Kabir, Chandra Shaker Chouhan.

**Supervision:** Mahbubul Pratik Siddique, A. K. M. Anisur Rahman, Md. Amimul Ehsan.

**Writing – original draft:** Ajran Kabir, Chandra Shaker Chouhan, Mahbubul Pratik Siddique.

**Writing – review & editing:** Mahbubul Pratik Siddique, A. K. M. Anisur Rahman, Azimun Nahar, Md. Siddiqur Rahman, Md. Amimul Ehsan.

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
