## [Decision Letter · Decision Letter 0]

27 May 2024

PONE-D-24-13222Epidemiology of Canine Ehrlichiosis and Molecular Characterization of Erhlichia Canis in Bangladeshi Pet DogsPLOS ONE

Dear Dr. Ehsan,

Thank you for submitting your manuscript to PLOS ONE. After careful consideration, we feel that it has merit but does not fully meet PLOS ONE’s publication criteria as it currently stands. Therefore, we invite you to submit a revised version of the manuscript that addresses the points raised during the review process.

**ACADEMIC EDITOR:**It is suitable for re-review after major comments.  ==============================

We look forward to receiving your revised manuscript.

Kind regards,

Faham Khamesipour, Ph.D.

Academic Editor

PLOS ONE

2. In your Methods section, please provide additional details regarding participant consent from the owners of the animals. In the ethics statement in the Methods and online submission information, please ensure that you have specified (1) whether consent was informed and (2) what type you obtained (for instance, written or verbal). If the need for consent was waived by the ethics committee, please include this information.

A clean copy of the edited manuscript (uploaded as the new *manuscript* file).

“This project was funded by Bangladesh Agricultural University Research System (BAURES) (Project No. 2018/557/AU-GC).”

“This project was funded by the Bangladesh Agricultural University Research System (BAURES) (Project No. 2018/557/AU-GC).”

“This project was funded by Bangladesh Agricultural University Research System (BAURES) (Project No. 2018/557/AU-GC).”

6. In the online submission form, you indicated that [Data can be shared only on request].

7. PLOS ONE now requires that authors provide the original uncropped and unadjusted images underlying all blot or gel results reported in a submission’s figures or Supporting Information files. This policy and the journal’s other requirements for blot/gel reporting and figure preparation are described in detail at https://journals.plos.org/plosone/s/figures#loc-blot-and-gel-reporting-requirements and https://journals.plos.org/plosone/s/figures#loc-preparing-figures-from-image-files. When you submit your revised manuscript, please ensure that your figures adhere fully to these guidelines and provide the original underlying images for all blot or gel data reported in your submission. See the following link for instructions on providing the original image data: https://journals.plos.org/plosone/s/figures#loc-original-images-for-blots-and-gels.  

8. We note that Figure 1 in your submission contain [map/satellite] images which may be copyrighted. All PLOS content is published under the Creative Commons Attribution License (CC BY 4.0), which means that the manuscript, images, and Supporting Information files will be freely available online, and any third party is permitted to access, download, copy, distribute, and use these materials in any way, even commercially, with proper attribution. For these reasons, we cannot publish previously copyrighted maps or satellite images created using proprietary data, such as Google software (Google Maps, Street View, and Earth). For more information, see our copyright guidelines: http://journals.plos.org/plosone/s/licenses-and-copyright.

Additional Editor Comments:

I want you to respond to the comments and revise the manuscript. Given the growing dog population in Bangladesh and the significant zoonotic implications of the disease, we may find merit in favorably considering this manuscript.

Reviewers' comments:

Reviewer's Responses to Questions

**Comments to the Author**

1. Is the manuscript technically sound, and do the data support the conclusions?

Reviewer #1: Yes

Reviewer #2: No

Reviewer #3: Yes

2. Has the statistical analysis been performed appropriately and rigorously? 

Reviewer #1: Yes

Reviewer #2: No

Reviewer #3: No

3. Have the authors made all data underlying the findings in their manuscript fully available?

Reviewer #1: No

Reviewer #2: No

Reviewer #3: No

4. Is the manuscript presented in an intelligible fashion and written in standard English?

Reviewer #1: Yes

Reviewer #2: No

Reviewer #3: Yes

5. Review Comments to the Author

Reviewer #1: The authors comprehensively explored the epidemiology of canine Ehrlichiosis (Ehrlichia canis), bolstering their insights with meticulous molecular and microscopic analyses. Their discoveries represent a significant advancement for the study region, shedding light on a topic often overlooked despite its zoonotic importance. I commend the authors for undertaking this vital research. Their manuscript is skillfully crafted and logically presented.

While the manuscript is commendable, there are some areas that could benefit from refinement:

1. Ensuring consistency in the presentation of scientific names, adhering to journal guidelines for formatting.

2. Correcting a typo in the methodology section regarding the abbreviation "EDTA" (line 132)

3. Standardizing the presentation of primer names and codon hyphenation (line 145).

4. Removing redundant terminology, such as "status," from line 165.

5. Adding punctuation for clarity in line 186 after mentioning the Rajshahi district's prevalence.

6. Inserting a comma after "Positive control" in line 192.

7. Streamlining Section 3.2 by summarizing significant findings rather than duplicating data from tables.

8. Providing high-resolution versions of figures for enhanced clarity, particularly Figure 1.

9. Clarifying the source of the positive control used in Figure 2, considering it's the first report of E. canis in the study area.

10. Enhancing Figure 4 by incorporating the 18 partial sequences obtained in the study to offer a comprehensive view of diversity.

11. Providing the questionnaire used in this study as a supplementary file

Reviewer #2: The article in question suffers from several methodological and conceptual flaws that compromise its suitability for publication. First, the lack of rigor in standardizing the writing of scientific terms, the lack of adequate references for key statements, the inadequate sample size, the non-random sampling, and the lack of justification for the choice of expected prevalence in the sample analysis are significant flaws that undermine the study's credibility.

Other problematic issues include a lack of clarity in the definition of the target population of dogs studied, as well as a sample collection interval that is too long for a cross-sectional study. In addition, there are conceptual inaccuracies, such as the misuse of the term "risk factor" in a cross-sectional study and statements about the sensitivity of diagnostic techniques that lack substantiation. Finally, the quality of the presented visual evidence is questionable, featuring poor-quality images and subjective interpretations. All considerations regarding the manuscript are listed below. Therefore, I recommend rejecting the article as submitted for publication in Plos One.

Major Revision

Line 112. To calculate the sample size, justify the use of an expected prevalence of 20%. It would be correct to use 50% of the expected prevalence. This is especially true in areas where there are no molecular epidemiologic studies to determine the prevalence of E. canis infection in dogs. With an expected prevalence of 50%, the authors would achieve the highest possible sample size with a precision of 5%.

Line 108. The criteria for determining the target population of dogs is not entirely clear. Did the study include both sick and healthy dogs, or did it only include sick dogs? Regardless of their living area, dogs exhibiting clinical signs are more likely to contract E. canis.

Line 124. Contrary to what the authors claim, the sampling was not random because the authors used the criterion of animals attending veterinary clinics to select the animals. This is an important selection bias since the dogs that visit the veterinary clinics do not adequately represent the same characteristics as the population of dogs that live in the rural and urban areas of the different districts of Bangladesh. Therefore, we cannot directly extrapolate the results of the analysis of the sample dogs to the dog population in Bangladesh.

Line 165. The text uses the term "risk factor" inappropriately throughout. It is not possible to analyze risk factors in cross-sectional epidemiologic studies because it is not possible in this type of study to determine the temporality between cause and effect, i.e., it is not possible to establish a precise time between exposure to associated factors and PCR analysis of E. canis in dogs.

Lines 168-169. The authors cannot state the mortality rate with absolute accuracy. It is not possible to guarantee that all dog owners have reported their dog's death.

Lines 167-168. The PCR is more sensitive than the blood smear test. Using this case definition significantly increases the number of false negative results. This introduces a measurement bias into the study, which will impact the analysis of factors associated with E. canis infection in dogs.

Lines 188-189. Ehrlichia canis is usually found in monocytes but rarely in neutrophils. The presence of E. canis morulae in lymphocytes is unlikely. The authors present an image (Figure 3) that lacks credibility and is of poor quality. The arrows indicate the presence of artifacts within the erythrocytes.

Minor Revision

Line 57: Write E. canis in full. Write scientific names in full at the beginning of a sentence.

Line 60. Write E. canis in full.

Lines 62-64. The etiologic agent (E. canis) is responsible for the disease. Provide a reference for this sentence.

Lines 64-66. The clinical manifestations of CME are not directly comparable to those of HME. HME typically manifests as a mild disease in the vast majority of cases, whereas immunocompromised individuals experience the severe form of the disease.

Lines 65-66. Add a citation for this sentence.

Line 67: Replace the term subclinical with subacute.

Line 71. E. canis in full

Lines 131-132. This sampling period is relatively long for a cross-sectional epidemiologic study. The authors should justify the sample collection interval and its implications for cross-sectional studies.

Line 132 EDTA

Lines 136-139. Explain how each variable was categorized.

Line 147: Ehrlichia in italics.

Reviewer #3: PONE-D-24-13222

The data presented in this manuscript are very relevant addressing a vector-borne disease in dogs prevalent in several countries, but some changes are recommended.

Title: Epidemiology of Canine Ehrlichiosis and Molecular Characterization of Erhlichia canis in Bangladeshi Pet Dogs

Change to lower case in the species.

• Introduction

“Ehrlichia canis (E. canis) is the most critical and deadly species among different members of the genus Ehrlichia, causing canine monocytic ehrlichiosis (CME), also known as tropical canine pancytopenia”.

Although E. canis can be fatal, especially in the chronic phase of the disease. I recommend changing this phrase, as it may convey the erroneous idea that this agent is more pathogenic than others of the same genus for different hosts.

• Material and Methods

When calculating the mortality rate, was the population at risk considered?

“Mortality: The estimated total number of deaths in a population of a given sex and/or age, divided by the total number of this population, expressed per 100,000 population, for a given year, in a given country, territory, or geographic area.”

(https://www.who.int/data/gho/indicator-metadata-registry/imr-details/1157)

• Discussion

“Notably, half of the dogs that contracted ehrlichiosis succumbed to the disease”.

Review the discussion.

As the sample was collected from veterinary clinics, and not through household surveys, there is a sampling bias. There is a tendency for dog owners to seek veterinary assistance when their dogs are sick, therefore, dogs that have mild signs or those that are in a subclinical phase may not have had the infection investigated. Furthermore, the mortality rate must consider the entire population at risk, not just those cared for in clinics.

• References

Standardize formatting

Improve the quality of Figure 3.

6. PLOS authors have the option to publish the peer review history of their article (what does this mean?). If published, this will include your full peer review and any attached files.

Reviewer #1: **Yes: **Jayedul Hassan

Reviewer #2: No

Reviewer #3: No

---

## [Author Response · Author response to Decision Letter 0]

28 Sep 2024

The authors are highly obliged and thankful to the reviewers for their constructive comments, which helped us improve the quality of the paper in terms of science and presentation. The changes we made according to the reviewers' comments are highlighted as track changes in the revised manuscript.

Reviewer 1: The authors comprehensively explored the epidemiology of canine Ehrlichiosis (Ehrlichia canis), bolstering their insights with meticulous molecular and microscopic analyses. Their discoveries represent a significant advancement for the study region, shedding light on a topic often overlooked despite its zoonotic importance. I commend the authors for undertaking this vital research. Their manuscript is skillfully crafted and logically presented.

While the manuscript is commendable, there are some areas that could benefit from refinement:

1. Ensuring consistency in the presentation of scientific names, adhering to journal guidelines for formatting.

Response: Thank you for your suggestion. We have ensured consistency in the presentation of scientific names throughout the manuscript, adhering strictly to the journal's guidelines for formatting.

2. Correcting a typo in the methodology section regarding the abbreviation "EDTA" (line 132)

Response: Thank you for pointing out the typo in the methodology section. The abbreviation "EDTA" on line 132 has been corrected.

3. Standardizing the presentation of primer names and codon hyphenation (line 145).

Response: We have standardized the presentation of primer names and ensured proper hyphenation of codons as per your suggestion on line 145.

4. Removing redundant terminology, such as "status," from line 165.

Response: We have removed the redundant term "status" from line 165.

5. Adding punctuation for clarity in line 186 after mentioning the Rajshahi district's prevalence.

Response: Thank you for your suggestion. We have added the necessary punctuation for clarity in line 186 after mentioning the Rajshahi district's prevalence.

6. Inserting a comma after "Positive control" in line 192.

Response: We have inserted a comma after "Positive control" in line 192.

7. Streamlining Section 3.2 by summarizing significant findings rather than duplicating data from tables.

Response: Thank you for your valuable feedback. We have streamlined Section 3.2 by summarizing the significant findings instead of duplicating data from the tables.

8. Providing high-resolution versions of figures for enhanced clarity, particularly Figure 1.

Response: We have provided high-resolution versions of the figures for enhanced clarity.

9. Clarifying the source of the positive control used in Figure 2, considering it's the first report of E. canis in the study area.

Response: We have clarified that commercial positive DNA was used as the positive control in Figure 2, and this information is mentioned in the manuscript.

10. Enhancing Figure 4 by incorporating the 18 partial sequences obtained in the study to offer a comprehensive view of diversity.

Response: We have enhanced Figure 4 by incorporating the 18 new partial sequences obtained in the study, providing a more comprehensive view of diversity.

11. Providing the questionnaire used in this study as a supplementary file

Response: The questionnaire used in this study has been provided as a supplementary file.

Reviewer 2: The article in question suffers from several methodological and conceptual flaws that compromise its suitability for publication. First, the lack of rigor in standardizing the writing of scientific terms, the lack of adequate references for key statements, the inadequate sample size, the non-random sampling, and the lack of justification for the choice of expected prevalence in the sample analysis are significant flaws that undermine the study's credibility.

Other problematic issues include a lack of clarity in the definition of the target population of dogs studied, as well as a sample collection interval that is too long for a cross-sectional study. In addition, there are conceptual inaccuracies, such as the misuse of the term "risk factor" in a cross-sectional study and statements about the sensitivity of diagnostic techniques that lack substantiation. Finally, the quality of the presented visual evidence is questionable, featuring poor-quality images and subjective interpretations. All considerations regarding the manuscript are listed below. Therefore, I recommend rejecting the article as submitted for publication in Plos One.

Major Revision

Line 112. To calculate the sample size, justify the use of an expected prevalence of 20%. It would be correct to use 50% of the expected prevalence. This is especially true in areas where there are no molecular epidemiologic studies to determine the prevalence of E. canis infection in dogs. With an expected prevalence of 50%, the authors would achieve the highest possible sample size with a precision of 5%.

Response: Thank you for your comment. We acknowledge that using 50% as the expected prevalence would have maximized the sample size, particularly in the absence of prior reports specific to our study area. However, we opted to base our expected prevalence on the average prevalence of Ehrlichiosis in South Asia determined through PCR, which was reported as 19.8%. We rounded this to 20% for simplicity and alignment with available literature (please check: https://doi.org/10.3390/vetsci10010021). 

Line 108. The criteria for determining the target population of dogs is not entirely clear. Did the study include both sick and healthy dogs, or did it only include sick dogs? Regardless of their living area, dogs exhibiting clinical signs are more likely to contract E. canis.

Response: In the revised manuscript, we have provided a clear description of the study and target populations. Our study included dogs irrespective of their health status and living area.

Line 124. Contrary to what the authors claim, the sampling was not random because the authors used the criterion of animals attending veterinary clinics to select the animals. This is an important selection bias since the dogs that visit the veterinary clinics do not adequately represent the same characteristics as the population of dogs that live in the rural and urban areas of the different districts of Bangladesh. Therefore, we cannot directly extrapolate the results of the analysis of the sample dogs to the dog population in Bangladesh.

Response: We agree that the sampling type was not random but purposive. We have corrected this in the revised manuscript. Pet dogs from both rural and urban areas commonly visit veterinary clinics for routine health check-ups, deworming, and vaccinations. Therefore, we can justify that the sample tested from various veterinary clinics represents the total dog population in Bangladesh.

Line 165. The text uses the term "risk factor" inappropriately throughout. It is not possible to analyze risk factors in cross-sectional epidemiologic studies because it is not possible in this type of study to determine the temporality between cause and effect, i.e., it is not possible to establish a precise time between exposure to associated factors and PCR analysis of E. canis in dogs.

Response: We acknowledge the concern raised regarding the use of the term "risk factor" in a cross-sectional study. Indeed, cross-sectional studies cannot establish temporality between exposure and outcome, which is crucial for determining causality. However, reporting associations between exposure and outcome in cross-sectional studies serves to generate hypotheses for further investigation in prospective studies, where temporality can be established. While cross-sectional studies are limited in their ability to establish causality, they are valuable for their ease of implementation, requiring less time and resources compared to cohort studies. They are also widely used in epidemiological research, providing important insights into potential associations and guiding future research directions.

Lines 168-169. The authors cannot state the mortality rate with absolute accuracy. It is not possible to guarantee that all dog owners have reported their dog's death.

Response: Thank you for your comment. We agree that it is challenging to ensure complete accuracy in reporting the mortality. However, since only 17 dogs were infected with Ehrlichiosis, we were able to easily communicate with their owners via telephone to ascertain the health status of these dogs. This allowed us to gather reasonably accurate information about their outcomes.

Lines 167-168. The PCR is more sensitive than the blood smear test. Using this case definition significantly increases the number of false negative results. This introduces a measurement bias into the study, which will impact the analysis of factors associated with E. canis infection in dogs.

Response: We agree with the reviewer that PCR is more sensitive than the blood smear test. To address this, we examined all 246 samples using both smear tests and PCR. Our case definition considered samples positive only if they tested positive in both methods. This approach minimizes the probability of false negatives. We have clarified this information in the revised manuscript.

Lines 188-189. Ehrlichia canis is usually found in monocytes but rarely in neutrophils. The presence of E. canis morulae in lymphocytes is unlikely. The authors present an image (Figure 3) that lacks credibility and is of poor quality. The arrows indicate the presence of artifacts within the erythrocytes.

Response: We appreciate the reviewer's feedback regarding Figure 3. Upon reevaluation, we acknowledge that the figure indeed did not capture any monocytes, and the quality was suboptimal, showing artifacts within the erythrocytes rather than definitive E. canis morulae. Consequently, we have decided to remove Figure 3 from the manuscript to maintain the integrity and quality of our presentation. Thank you for bringing this to our attention.

Minor Revision

Line 57: Write E. canis in full. Write scientific names in full at the beginning of a sentence.

Response: Thank you for your suggestion. We have corrected the manuscript to write Ehrlichia canis in full, especially at the beginning of a sentence, as recommended.

Line 60. Write E. canis in full.

Response: Corrected accordingly.

Lines 62-64. The etiologic agent (E. canis) is responsible for the disease. Provide a reference for this sentence.

Response: Provided according to your comment

Lines 64-66. The clinical manifestations of CME are not directly comparable to those of HME. HME typically manifests as a mild disease in the vast majority of cases, whereas immunocompromised individuals experience the severe form of the disease.

Response: We have modified these statements as per your comment.

Lines 65-66. Add a citation for this sentence.

Response: We have added a citation according to your comment.

Line 67: Replace the term subclinical with subacute.

Response: We have replaced subclinical with subacute.

Line 71. E. canis in full

Response: Corrected accordingly.

Lines 131-132. This sampling period is relatively long for a cross-sectional epidemiologic study. The authors should justify the sample collection interval and its implications for cross-sectional studies.

Response: Thank you for your comment. The sample collection period spanned a relatively long interval, including the duration of the COVID-19 pandemic. This extended period was necessary due to logistical challenges posed by the pandemic. However, since there are no ongoing control or preventive measures for Ehrlichiosis in the study area, the long duration of the study is unlikely to significantly impact the findings.

Line132: EDTA

Response: Corrected

Lines 136-139. Explain how each variable was categorized.

Response: We have provided a detailed explanation of how each variable was categorized in the revised manuscript.

Line 147: Ehrlichia in italics.

Response: Done as suggested.

Reviewer 3: 

The data presented in this manuscript are very relevant addressing a vector-borne disease in dogs prevalent in several countries, but some changes are recommended.

Title: Epidemiology of Canine Ehrlichiosis and Molecular Characterization of Erhlichia canis in Bangladeshi Pet Dogs

Change to lower case in the species.

Response: We have changed it.

• Introduction

“Ehrlichia canis (E. canis) is the most critical and deadly species among different members of the genus Ehrlichia, causing canine monocytic ehrlichiosis (CME), also known as tropical canine pancytopenia”. Although E. canis can be fatal, especially in the chronic phase of the disease. I recommend changing this phrase, as it may convey the erroneous idea that this agent is more pathogenic than others of the same genus for different hosts.

Response: We have rephrased it as: Ehrlichia canis (E. canis) is one of the most critical and deadly species within the genus Ehrlichia. It causes canine monocytic ehrlichiosis (CME), also known as tropical canine pancytopenia.

•Material and Methods

When calculating the mortality rate, was the population at risk considered? “Mortality: The estimated total number of deaths in a population of a given sex and/or age, divided by the total number of this population, expressed per 100,000 population, for a given year, in a given country, territory, or geographic area.”(https://www.who.int/data/gho/indicator-metadata-registry/imr-details/1157)

Response: Thank you for your question. Based on the reviewer's subsequent comments, we have removed the mortality risk from the table and its description.

• Discussion

“Notably, half of the dogs that contracted ehrlichiosis succumbed to the disease”. Review the discussion. As the sample was collected from veterinary clinics, and not through household surveys, there is a sampling bias. There is a tendency for dog owners to seek veterinary assistance when their dogs are sick, therefore, dogs that have mild signs or those that are in a subclinical phase may not have had the infection investigated. Furthermore, the mortality rate must consider the entire population at risk, not just those cared for in clinics.

Response: We acknowledge the reviewer's concerns about potential sampling bias in our study, as the sample was collected from veterinary clinics rather than through household surveys. It is true that dog owners tend to seek veterinary assistance when their dogs are sick, which may lead to an overrepresentation of dogs with more severe clinical signs, while those with mild signs or in a subclinical phase might not have been investigated. However, it is also important to note that dogs are brought to veterinary clinics not only for illness but also for deworming, vaccination, and routine health check-ups. The dog owners in our sample are generally educated and affluent, representing a segment of society that is particularly aware of their pets' health. Therefore, we believe our sample provides a representative snapshot of the pet dog population in Bangladesh. We recognize this as a limitation of our study, as it only includes dogs that visited pet clinics. To explore the full burden of the disease, future studies should also consider stray dogs and those not routinely brought to clinics. We have discussed this in the revised manuscript. We have deleted mortality from table and description.

• References

Standardize formatting

Response: We have formatted the reference list.

Improve the quality of Figure 3.

Response: We have deleted Figure 3.

---

## [Decision Letter · Decision Letter 1]

15 Oct 2024

PONE-D-24-13222R1Epidemiology of Canine Ehrlichiosis and Molecular Characterization of Erhlichia Canis in Bangladeshi Pet DogsPLOS ONE

Dear Dr. Ehsan,

Thank you for submitting your manuscript to PLOS ONE. After careful consideration, we feel that it has merit but does not fully meet PLOS ONE’s publication criteria as it currently stands. Therefore, we invite you to submit a revised version of the manuscript that addresses the points raised during the review process.

**ACADEMIC EDITOR:**Minor revisions are needed. I want you to respond to the comments and revise the manuscript. ==============================

We look forward to receiving your revised manuscript.

Kind regards,

Faham Khamesipour, Ph.D.

Academic Editor

PLOS ONE

Journal Requirements:

Additional Editor Comments:

Minor revisions are needed. I want you to respond to the comments and revise the manuscript.

Reviewers' comments:

Reviewer's Responses to Questions

**Comments to the Author**

1. If the authors have adequately addressed your comments raised in a previous round of review and you feel that this manuscript is now acceptable for publication, you may indicate that here to bypass the “Comments to the Author” section, enter your conflict of interest statement in the “Confidential to Editor” section, and submit your "Accept" recommendation.

Reviewer #1: All comments have been addressed

Reviewer #3: All comments have been addressed

2. Is the manuscript technically sound, and do the data support the conclusions?

Reviewer #1: Yes

Reviewer #3: Yes

3. Has the statistical analysis been performed appropriately and rigorously? 

Reviewer #1: Yes

Reviewer #3: No

4. Have the authors made all data underlying the findings in their manuscript fully available?

Reviewer #1: Yes

Reviewer #3: Yes

5. Is the manuscript presented in an intelligible fashion and written in standard English?

Reviewer #1: Yes

Reviewer #3: Yes

6. Review Comments to the Author

Reviewer #1: I appreciate your response and the thoughtful way the comments were addressed. However, the supplementary materials were not cited in the main text. I recommend that the authors cite the supplementary files "Supplementary 1.docx" and "Erlichia Supplementary.xlsx" in the appropriate sections of the main text. Additionally, please remove "Supporting Information Fig. 2 raw.tif.

Reviewer #3: Line 54 - "E. canis", scientific genus name at the beginning of a sentence should be written in full.

Line 93 – correct “… in south a sian countries…”

Item 2.8 Statistical analysis also contains mortality

“…The mortality and case fatality was calculated based on the follow-up communication with the dog owners. Z-tests for proportions were used to compare prevalence, mortality, and case fatality among districts [24, 25].

Format tables according to the journal's standards. For example, Table 2 has rows without data

7. PLOS authors have the option to publish the peer review history of their article (what does this mean?). If published, this will include your full peer review and any attached files.

Reviewer #1: **Yes: **Jayedul Hassan

Reviewer #3: No

---

## [Author Response · Author response to Decision Letter 1]

17 Oct 2024

The authors sincerely thank the reviewers for their constructive comments, which greatly improved the scientific content and presentation of the paper. All changes made in response to their suggestions are highlighted as track changes in the revised manuscript.

Reviewer #1: I appreciate your response and the thoughtful way the comments were addressed. However, the supplementary materials were not cited in the main text. I recommend that the authors cite the supplementary files "Supplementary 1.docx" and "Erlichia Supplementary.xlsx" in the appropriate sections of the main text. Additionally, please remove "Supporting Information Fig. 2 raw.tif.

Response: Thank you for your valuable feedback. We have cited the supplementary files ("Supplementary 1.docx" and "Erlichia Supplementary.xlsx") in the relevant sections of the main text as suggested, and we have also removed "Supporting Information Fig. 2 raw.tif" as recommended.names throughout the manuscript, adhering strictly to the journal’s guidelines for formatting.

Reviewer #3: Line 54 - "E. canis", scientific genus name at the beginning of a sentence should be written in full.

Response: Thank you for bringing this to our attention. We've gone through the manuscript and made sure that "E. canis" is written in full as Ehrlichia canis at the beginning of sentences.

Line 93 – correct “… in south a sian countries…”

Response: Corrected

Item 2.8 Statistical analysis also contains mortality

“…The mortality and case fatality was calculated based on the follow-up communication with the dog owners. Z-tests for proportions were used to compare prevalence, mortality, and case fatality among districts [24, 25].

Response: Thank you for your observation. We have made the necessary corrections in the manuscript

Format tables according to the journal's standards. For example, Table 2 has rows without data

Response: Thank you for your feedback. We have reformatted the tables to meet the journal's standards and removed any empty rows from Table 2.

---

## [Decision Letter · Decision Letter 2]

4 Nov 2024

PONE-D-24-13222R2Epidemiology of Canine Ehrlichiosis and Molecular Characterization of Erhlichia Canis in Bangladeshi Pet DogsPLOS ONE

Dear Dr. Ehsan,

Thank you for submitting your manuscript to PLOS ONE. After careful consideration, we feel that it has merit but does not fully meet PLOS ONE’s publication criteria as it currently stands. Therefore, we invite you to submit a revised version of the manuscript that addresses the points raised during the review process.

**ACADEMIC EDITOR**:Minor revisions are needed. The authors have made the suggested changes, but should carefully review the entire text, as it still contains information about mortality. I want you to respond to the comments and revise the manuscript.==============================

We look forward to receiving your revised manuscript.

Kind regards,

Faham Khamesipour, Ph.D.

Academic Editor

PLOS ONE

Journal Requirements:

Additional Editor Comment:

Minor revisions are needed. The authors have made the suggested changes, but should carefully review the entire text, as it still contains information about mortality. I want you to respond to the comments and revise the manuscript.

Reviewers' comments:

Reviewer's Responses to Questions

**Comments to the Author**

1. If the authors have adequately addressed your comments raised in a previous round of review and you feel that this manuscript is now acceptable for publication, you may indicate that here to bypass the “Comments to the Author” section, enter your conflict of interest statement in the “Confidential to Editor” section, and submit your "Accept" recommendation.

Reviewer #1: All comments have been addressed

Reviewer #3: All comments have been addressed

2. Is the manuscript technically sound, and do the data support the conclusions?

Reviewer #1: Yes

Reviewer #3: Yes

3. Has the statistical analysis been performed appropriately and rigorously? 

Reviewer #1: Yes

Reviewer #3: Yes

4. Have the authors made all data underlying the findings in their manuscript fully available?

Reviewer #1: Yes

Reviewer #3: Yes

5. Is the manuscript presented in an intelligible fashion and written in standard English?

Reviewer #1: Yes

Reviewer #3: Yes

6. Review Comments to the Author

Reviewer #1: The authors are highly appreciated for making modifications as suggested. I believe this study will contribute combating the deadliest disease in the study area and pave the way to further research.

Reviewer #3: The authors have made the suggested changes, but should carefully review the entire text, as it still contains information about mortality. For example: line 252 "The present study's overall mortality and case fatality were 3.3 % and 47.1%, respectively"

7. PLOS authors have the option to publish the peer review history of their article (what does this mean?). If published, this will include your full peer review and any attached files.

Reviewer #1: **Yes: **Jayedul Hassan

Reviewer #3: No

---

## [Author Response · Author response to Decision Letter 2]

7 Nov 2024

We sincerely thank the reviewers for their valuable feedback and for identifying areas needing further attention. The changes we made in response to their comments are highlighted as tracked changes in the revised manuscript.

Reviewer #3: The authors have made the suggested changes, but should carefully review the entire text, as it still contains information about mortality. For example: line 252 "The present study's overall mortality and case fatality were 3.3 % and 47.1%, respectively"

Response: Thank you for your feedback. We have carefully reviewed the entire text to ensure that all references to mortality statistics have been removed from the manuscript. Please see lines 252–261 for the relevant changes.

---

## [Decision Letter · Decision Letter 3]

15 Nov 2024

Epidemiology of Canine Ehrlichiosis and Molecular Characterization of Erhlichia Canis in Bangladeshi Pet Dogs

PONE-D-24-13222R3

Dear Dr. Ehsan,

We’re pleased to inform you that your manuscript has been judged scientifically suitable for publication and will be formally accepted for publication once it meets all outstanding technical requirements.

Kind regards,

Faham Khamesipour, Ph.D.

Academic Editor

PLOS ONE

Additional Editor Comments (optional):

Reviewers' comments:

Reviewer's Responses to Questions

**Comments to the Author**

1. If the authors have adequately addressed your comments raised in a previous round of review and you feel that this manuscript is now acceptable for publication, you may indicate that here to bypass the “Comments to the Author” section, enter your conflict of interest statement in the “Confidential to Editor” section, and submit your "Accept" recommendation.

Reviewer #1: All comments have been addressed

Reviewer #3: All comments have been addressed

2. Is the manuscript technically sound, and do the data support the conclusions?

Reviewer #1: Yes

Reviewer #3: Yes

3. Has the statistical analysis been performed appropriately and rigorously? 

Reviewer #1: Yes

Reviewer #3: Yes

4. Have the authors made all data underlying the findings in their manuscript fully available?

Reviewer #1: Yes

Reviewer #3: Yes

5. Is the manuscript presented in an intelligible fashion and written in standard English?

Reviewer #1: Yes

Reviewer #3: Yes

6. Review Comments to the Author

Reviewer #1: (No Response)

Reviewer #3: (No Response)

7. PLOS authors have the option to publish the peer review history of their article (what does this mean?). If published, this will include your full peer review and any attached files.

Reviewer #1: **Yes: **Jayedul Hassan

Reviewer #3: No

---

## [Editor Report · Acceptance letter]

25 Nov 2024

PONE-D-24-13222R3 

PLOS ONE

Dear Dr. Ehsan, 

I'm pleased to inform you that your manuscript has been deemed suitable for publication in PLOS ONE. Congratulations! Your manuscript is now being handed over to our production team.

Kind regards, 

on behalf of

Dr. Faham Khamesipour 

Academic Editor

PLOS ONE